# Emotion Recognition in HMDs: A Multi-task Approach Using Physiological Signals and Occluded Faces

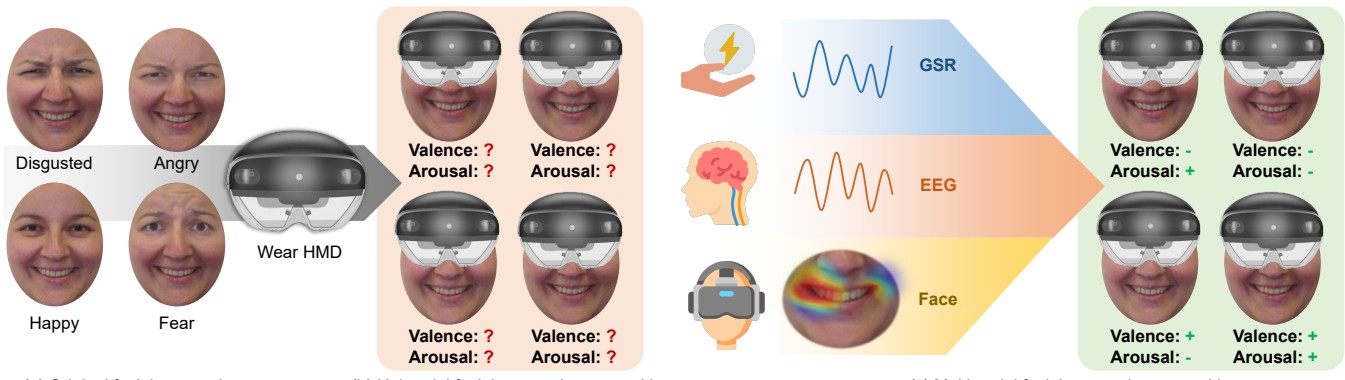

**Figure 1: The examples of difficulty in facial expression recognition under partial occlusion that human in (a) after wearing a Head-mounted displays (HMDs) to become (b). (c) shows the multimodal fusion between occluded facial and physiological information. The photographs of human are from the Karolinska Directed Emotional Faces [28].**

## ABSTRACT

Prior research on emotion recognition in extended reality (XR) has faced challenges due to the occlusion of facial expressions by Head-Mounted Displays (HMDs). This limitation hinders accurate Facial Expression Recognition (FER), which is crucial for immersive user experiences. This study aims to overcome the occlusion challenge by integrating physiological signals with partially visible facial expressions to enhance emotion recognition in XR environments. We employed a multi-task approach, utilizing a feature-level fusion to fuse Electroencephalography (EEG) and Galvanic Skin Response (GSR) signals with occluded facial expressions. The model predicts valence and arousal simultaneously from both macro-and micro-expression. Our method demonstrated improved accuracy in emotion recognition under partial occlusion conditions. The integration of temporal physiological signals with other modalities significantly enhanced performance, particularly for half-face emotion recognition. The study presents a novel approach to emotion recognition in XR, addressing the limitations of facial occlusion by HMDs. The findings suggest that physiological signals are vital for interpreting emotions in occluded scenarios, offering potential for real-time applications and advancing social XR applications.

*ACM MM, 2024, Melbourne, Australia*
© 2024 Copyright held by the owner/author(s). Publication rights licensed to ACM.
ACM ISBN 978-x-xxxx-xxxx-x/YY/MM
https://doi.org/10.1145/nnnnnnn.nnnnnnn

## CCS CONCEPTS

• **Human-centered computing → Interaction paradigms**.

## KEYWORDS

Head-mounted Displays, Extended Reality, Valence, Arousal, Emotion, Multi-modal, Multi-task Learning

## 1 INTRODUCTION

Head-mounted displays (HMDs) are essential for experiencing extended reality (XR), which includes Augmented Reality (AR), Virtual Reality (VR), and Mixed Reality (MR). In this context, people's emotions are naturally and immediately stimulated due to the high immersion and realism of XR. For instance, Virtual Reality Therapy studies human emotional reactions to explore treatments for mental health disorders such as phobic disorders [18], obsessive-compulsive disorder [41], and eating disorders [6], or for social cognition training [31]. Moreover, VR content is adapted to the user's emotional state for relaxation in the virtual world [14]. However, HMDs pose a challenge to external recording techniques as they cover the user's upper face [11] (see Figure 1 (b)). This limitation greatly affects social XR applications, especially Facial Expression Recognition (FER), a key method for studying and differentiating human emotions. In these applications, facial features play a critical role in creating an immersive user experience [5]. The accuracy of FER decreases if the upper face information is discarded [5]. Another difficulty in FER in an immersive XR context is that conventional FER methods rely on public databases, which contain complete facial information. To adapt these datasets to the XR context, some studies have made adjustments, such as placing a "VR

patch" on a detected face [15, 33] and processing images of people wearing VR headsets into their corresponding Grad-CAM [37] explanation masks [10].

Apart from the issue of HMDs obscuring the face, the brief duration of facial expressions, whether they are facial macro-expressions that are voluntary muscle movements covering a large area of the face and lasting between 0.5 and 4 seconds [9], or facial micro-expressions that are brief, involuntary facial changes like nose wrinkling over a short time frame of between 65 and 500 ms [45], and their subtle movements make them difficult for humans to identify [36]. Human facial expressions may be a mix of facial micro-expressions and macro-expressions. For example, both the surprise from micro-expression and macro-expression involve raised eyebrows and opened eyes [44]. Micro-expressions are almost impossible to fake [39]. On the contrary, macro-expressions may not convey hidden emotions that determine true human feelings and state-of-mind [44], and can be easily controlled and manipulated.

Physiological responses, which are difficult to fake, provide a deeper understanding of underlying emotions [36]. These responses originate from the central (brain stem) and autonomic nervous systems, which regulate body functions like heart rate, respiration, blood pressure, swallowing, and pupillary movement [22]. Electroencephalography (EEG) can effectively reflect the electrical activity of the central nervous system and is associated with emotion [1, 25]. EEG is commonly used in emotional recognition studies in immersive virtual environments [47]. Peripheral physiological signals such as Galvanic Skin Response (GSR), photoplethysmogram (PPG), or Heart Rate Variability (HRV) can also be used to reliably measure emotional state and have been widely used in emotional engagement studies in immersive virtual environments [3, 16]. Although physiological signals cannot be deliberately controlled or concealed [43], they are suitable for real-time emotion recognition [27]. However, these signals can be very weak and easily contaminated by noise like artifacts [40].

Thus, recognizing emotions using only physiological signals or incomplete facial expressions can be quite challenging.

A few studies have used incomplete facial expressions and physiological signals instead of only incomplete facial expressions [10], but this area still requires further research and exploration. This paper combines various modalities (incomplete facial expressions and physiological signals) to address the limitations of each individual modality.

The contributions of this paper include:

- **Emotion Recognition in XR Scenarios**. We built a multi-task model for emotion recognition in XR scenarios. This model integrates physiological signals (EEG and GSR) and partial facial expressions to simultaneously predict both valence and arousal.
- **Integration of Temporal Physiological Signals**. Our research indicates that physiological signals enhance emotion recognition with occluded faces, especially for micro-expressions. Moreover, by concentrating on the lower face during training, we enhanced performance.
- **Validation on Datasets and Real-World Scenarios**. We tested our multi-modal, multi-task method on DEAP, AMI-GOS datasets, and real-world situations. Results show that

multi-tasking boosts accuracy and reduces prediction time, proving its effectiveness for HMD-based emotion recognition tasks.

## 2 RELATED WORK

### 2.1 Discrete and Dimensional Emotion Model

Emotion models are primarily divided into two categories: discrete and dimensional [13]. The discrete model includes six universally recognized emotions: anger, happiness, fear, surprise, disgust, and sadness [7, 8]. However, Saffaryazdi et al. [36] argue that categorizing emotional states into discrete emotions is incorrect as human emotions are often a blend of several feelings. For instance, reported happiness may be a mix of positive feelings. In situations of positive and negative smiles, brain patterns and physiological signals differ, leading to misclassification when grouped into a single class. The dimensional model, in contrast, views emotions as a combination of three psychological dimensions: arousal, valence, and either dominance or intensity [36]. Russel's Circumplex Model, proposed by Posner et al. [34], uses only valence and arousal to represent emotions, with valence indicating a range from negative to positive emotions, and arousal representing a spectrum from passive to active emotions. Most neurophysiological emotion recognition research [32] and benchmark datasets use the Circumplex Model for assessment. Our study uses the Circumplex Model to evaluate our method of detecting emotions using neurophysiological cues and facial expressions on two multimodal datasets.

### 2.2 Multimodal Datasets for Emotion Recognition

Multimodal datasets for emotion recognition have garnered significant interest among researchers [36]. A few such datasets, including facial video, EEG, and physiological signals, are available. The AMI-GOS [21] and DEAP [30] datasets are popular for macro-expression and micro-expression studies, respectively.

**Facial macro-expression dataset**. The AMIGOS dataset [21] includes neuro-physiological signals (EEG, ECG, GSR), frontal HD video, and full-body RGB and depth videos from 37 participants. It uses a variety of videos to elicit affective responses in individual and group settings. The EEG data, collected using a state-of-the-art device, along with self-assessment manikins (SAM) [2], provide a comprehensive study of affective responses.

**Facial micro-expression dataset**. The DEAP dataset [30] contains neuro-physiological signals (EEG, peripheral physiological signals) from 22 participants watching music video excerpts. It uses a wide range of music videos to induce affective responses. The EEG data, collected using a cutting-edge device, and participant ratings on arousal, valence, like/dislike, dominance, and familiarity, provide an extensive analysis of affective responses.

### 2.3 The Relationship and Fusion Between Modalities

**Relationship between modalities**. Some studies have explored the relationship between behavioral responses and physiological changes in multimodal emotion recognition. Saffaryazdi [36] suggested using facial micro-expressions in conjunction with brain

 

**Table 1: Uni/Multi-modal Full/Half Face Micro-/Macro-expression Recognition. Note: Partially occluded face (POFace), full face (FFace) macro-expression (MaE), Micro-expression (ME), dual-eye infrared images (DeIIs), functional near-infrared spectroscopy (fNIRS), electroencephalography (EEG), photoplethysmogram (PPG), and galvanic skin response (GSR).**

| Author | Modalities | Emotion Model | MaE/ME Dataset | Multimodal Fusion Method | Single/Multi-task |
|---|---|---|---|---|---|
| Houshmand et al. [15] | POFace | Discrete | MaE: FER+, AffectNet, RAF-DB | - | Single-task |
| Georgescu et al. [10] | POFace | Discrete | MaE: FER+, AffectNet | - | Single-task |
| Chen et al. [5] | POFace, DeIIs | Discrete | MaE: Custom dataset | Concatenation (feature level) | Single-task |
| Sun et al. [38] | fNIRS, EEG | Discrete | MaE: Custom dataset | Concatenation (feature level) | Single-task |
| Saffaryazdi et al. [36] | FFace, EEG, PPG, and GSR | Dimensional | ME: DEAP, MaE: Custom dataset | Vote fusion (decision level) | Single-task |
| Ours | POFace, EEG, and GSR | Dimensional | MaE: AMIGOS, ME: DEAP | Cross-attention (feature level) | Multi-task |

and physiological signals for more reliable detection of underlying emotions. Sun et al. [38] found a strong correlation in emotional valence between spontaneous facial expressions and brain activities measured by EEG. Liu et al. [26] discovered that EEG signals, reflecting brain activity, and GSR signals, linked to the autonomic nervous system, offer unique benefits in emotion recognition.

**Fusion between modalities**. Some methods fuse modalities at the feature level. Sun et al. [38] extracted discrete features from functional near-infrared spectroscopy (fNIRS) and electroencephalography (EEG), and simply concatenated these discrete features from both modalities for fusion. Zhang et al. [48] used a cross-attention mechanism to align the relationship between PPG and GSR modalities, highlighted important information within a single modality using a self-attention mechanism, and finally used a predictor to perform stress recognition on the representations of both modalities. Pan et al. [32] proposed an online emotion recognition method based on multimodal physiological signals (such as ECG, EEG, GSR, etc.), using hypergraph learning techniques to fuse multimodal information and capture complex and nonlinear relationships between data. Other methods fuse modalities at the decision level. Saffaryazdi et al. [36] fed preprocessed sequences into two 3D CNN models to extract features for classifying arousal and valence separately, used a Dense layer with Adam optimizer [20] to classify the physiological data for arousal and valence labels separately, and fused the classification results at the decision level.

## 2.4 Facial Expression Recognition Methods in XR Context

**Emotion recognition on occluded faces**. Current half-face emotion recognition is mainly divided into two categories. One uses the half-face as input for emotion modeling, which requires a simple network structure and has low training time overhead. However, due to the absence of important information from the upper face, such as eyes, eyebrows, and nose [5], the recognition accuracy is lower [10, 15]. The other category uses additional devices to supplement the occluded facial information, which helps improve recognition accuracy [5]. However, additional devices, such as cameras, may disrupt immersion and increase head load. Table 1 presents the related work on Uni/Multi-modal Full/Half Face Micro-/Macro-expression Recognition.

**Face reconstruction on occluded faces**. Some work uses methods like generative adversarial network (GAN) to "repair" the occluded part. For example, Zhao et al. [50] collected pre-recorded

sequences to reconstruct a 3D head, and Wang et al. [42] used GAN to repair incomplete facial information. These works may help achieve higher emotion recognition accuracy. However, these methods may not meet the real-time or low-latency requirements of XR (Mixed Reality) [24].

**Single-task learning vs multi-task learning**. Emotion recognition tasks for HMD XR applications may require real-time processing of multiple tasks, such as simultaneously predicting valence, arousal, etc. [4, 35]. Compared to training models for each task separately, building a multi-task learning model to simultaneously predict valence and arousal is more efficient. For example, Priyasad et al. [35] showed that multi-task learning can achieve fewer parameters, faster convergence, and higher performance for emotion recognition with EEG signals. Similarly, the multi-modal multi-task (M3T) approach [49] fused both visual features from facial videos and acoustic features from the audio tracks to estimate the valence and arousal simultaneously.

## 3 METHODOLOGY

### 3.1 Data Processing Module

#### 3.1.1 Samples with both Face Videos and Physical Signals.

**AMIGOS Dataset**. Out of 40 participants, 37 (excluding participants 8, 24, and 28) took part in both short and long video experiments designed to evoke emotional responses. Their facial videos, EEG, and peripheral physiological signals were fully recorded.

**DEAP dataset**. For the first 22 out of 32 participants, complete facial videos, EEG, and peripheral physiological signals were preserved. However, due to technical issues, some of the last trials for participants 3, 5, 11, and 14 are missing.

#### 3.1.2 Face Video Processing.

**Segmentation based on annotations**. In the AMIGOS dataset, videos capturing participants' faces were divided into multiple 20-second segments. The first 5 seconds before the stimuli presentation were extracted as the initial clip. Subsequently, starting from the 5-second mark (when the stimuli began), non-overlapping 20-second segments were extracted, with the number of segments depending on the duration of the stimuli video. The final clip included the last segment of the video, which was less than 20 seconds. To ensure uniform segment length, the first and last segments shorter than 20 seconds were discarded. Three annotators provided external annotations of valence and arousal for the 20-second segments of the face videos in both experiments. In the DEAP dataset, each face

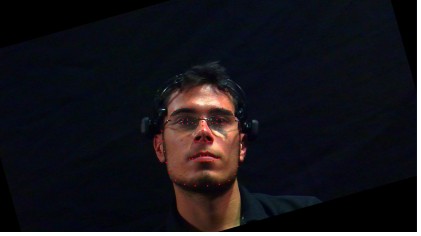
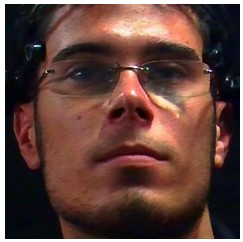
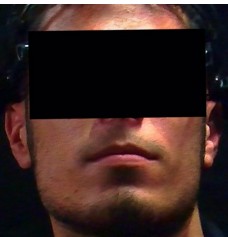

(a) Obtain 68 facial landmarks and 10 region blocks    (b) Alignment    (c) Cropping    (d) Artificial Occlusion

**Figure 2: The face video processing.**

video lasted approximately 60 seconds and was not segmented due to only one annotation within that duration.

**Apex frame finding**. We used an open-source facial toolkit[1] to obtain 68 facial landmarks (red points) in each frame of the facial expression sequence (See Figure 2(a)). However, due to factors like poor video quality and dim lighting, faces were not detected in some videos, with successful detection in approximately 70% of the videos. Following previous research, we defined 10 regions on the face based on the detected landmarks (yellow boxes), representing areas where muscle movements frequently occur. The size of each region was estimated to be half the width of the mouth. These regions are depicted in Figure 2(a). A facial video sequence starts with the Onset frame (when muscle movements occur in response to emotional stimuli) and ends with the Offset frame (when emotional reactions disappear and the face returns to a relaxed state). The Apex frame is the one with the most significant change in intensity within the sequence. To identify the apex frame, we calculated the absolute pixel differences between the current frame and the onset and offset frames in the ten regions. To minimize environmental noise, we normalized the sum of the differences by dividing it by the difference between the considered frame and its consecutive frame. We then obtained the per-pixel average value for each frame in the micro-expression (ME) sequence. The apex frame should indicate the peak of intensity differences with the onset and offset frames. Thus, we selected the frame with the highest per-pixel value of intensity differences as the apex frame:

$$S_i = f(\text{Frame}_i, \text{Frame}_{\text{onset}}) + f(\text{Frame}_i, \text{Frame}_{\text{offset}})$$
$$Index_{apex} = i_{max} = \underset{i}{\arg\max}(S_i) \qquad (1)$$

where

$$f(\text{Frame}_n, \text{Frame}_m) = \frac{|\text{Frame}_n - \text{Frame}_m| + 1}{|\text{Frame}_n - \text{Frame}_{n-\epsilon}| + 1} \qquad (2)$$

Function ($f$) measures the absolute pixel differences between two frames, normalized by the difference between the current frame and its third preceding frame to reduce noise. We calculate the absolute pixel differences in 10 region blocks generated by facial toolkit between the current frame and the onset frame, and the offset frame, respectively. To reduce the impact of environmental noise, divide the difference by the absolute pixel difference between the current frame and the continuous frame with a deviation of $\epsilon$. Sum the normalized pixel differences to represent the change

[1]https://github.com/ageitgey/face recognition/

intensity value of the current frame, denoted as $S_i$. $i_{max}$ identifies the apex frame showing the peak intensity of the facial expression. This frame is crucial for facial expression recognition tasks.

### 3.1.3 Finding the Region of Interest (ROI).

Considering the two datasets have frame rates of 25fps and 50fps, we reduced any video with a frame rate above 25fps to 25fps for consistency (100 frames collected in 4s). We then selected 100 frames (4s) surrounding the peak frame as the region of interest (ROI). If the peak frame is at least 2s away from the video segment's edge, we included 2s (50 frames) on either side of it. But if the peak frame is less than 2s from any edge, we added frames from the opposite side until we had a total of 100 frames.

### 3.1.4 Alignment and Cropping.

After detecting facial key points, we align the face by rotating the image based on the eye key points [19] (See Figure 2(b)). For example, we calculate the center coordinates of the left and right eyes (($x_{\text{left}}, y_{\text{left}}$) and ($x_{\text{right}}, y_{\text{right}}$)), and compute the angle $\theta$ between the line connecting the centers of the left and right eyes and the horizontal direction ($\arctan\left(\frac{y_{\text{right}} - y_{\text{left}}}{x_{\text{right}} - x_{\text{left}}}\right)$). We then rotate the image counterclockwise by $\theta$ with the center coordinates of the two eyes as the base point.

After the image is rotated, the landmark coordinates in the image also need to be rotated accordingly so that the landmark can match the rotated image. The rotation transformation can be represented as:

$$\begin{bmatrix} x' \\ y' \end{bmatrix} = \begin{bmatrix} \cos(\theta) & -\sin(\theta) \\ \sin(\theta) & \cos(\theta) \end{bmatrix} \begin{bmatrix} x - x_{\text{center}} \\ y - y_{\text{center}} \end{bmatrix} + \begin{bmatrix} x_{\text{center}} \\ y_{\text{center}} \end{bmatrix} \qquad (3)$$

where ($x', y'$) are the coordinates after rotation, ($x, y$) are the original coordinates, and ($x_{\text{center}}, y_{\text{center}}$) are the center coordinates of the two eyes.

After aligning the face, we crop the face to a fixed size based on the landmark [19]. For instance, the vertical direction is divided into three parts: the middle part, which is the pixel distance from the center of the two eye landmarks to the center of the mouth landmark, accounts for 35% of the vertical direction of the cropped face image; the bottom part accounts for 35%; and the top part accounts for 30%. The horizontal direction is centered on the midpoint of the leftmost and rightmost landmarks. After cropping the face, the size becomes $320 \times 320$.

We define a landmark transformation function. Since the image is cropped, the landmark coordinates need to be transformed again. The transformation can be represented as:

$$\begin{bmatrix} x' \\ y' \end{bmatrix} = \begin{bmatrix} x - x_{\text{crop}} \\ y - y_{\text{crop}} \end{bmatrix} \tag{4}$$

where $(x', y')$ are the coordinates after cropping, $(x, y)$ are the original coordinates, and $(x_{\text{crop}}, y_{\text{crop}})$ are the top-left coordinates of the cropping rectangle. After transforming the landmark coordinates, we obtain aligned face images.

### 3.1.5 Artificial Occlusion.

In an extended reality setting, the upper face is partially occluded by a Head-Mounted Display (HMD). Since there are no standard occlusion face image databases containing persons wearing HMDs, we established XR-occluded images by masking the upper region on the standard facial micro- and macro-expression images, based on the "VR patch" method [15]. We simulated facial occlusion based on the 68 landmarks detected in the apex frame. We initialized XR dimensions to an aspect ratio of approximately 2:1, based on the Hololens 2 headset. To uniformly scale the XR patch on the training images, we used the distance between the two temporal bones of the facial landmarks as a reference length. We then generated the polygonal occluding patch by setting the midpoint of the line passing through the eye center points as the center coordinate of the XR headset. To account for face rotations, we aligned the resized patch with the axis running through the eye centers. We obtained the angle of incline by determining the inverse tangent function of changes in y-coordinates to changes in x-coordinates of eye center points, and then we utilized the rotation matrix to rotate corner points of the blocking patch about its central pivot point on the coordinate plane accordingly. This geometric model provides a more realistic occlusion resulting from wearing an XR headset, rather than simply covering the upper half of the face, as done in [15]. The same operation was performed for both datasets.

### 3.1.6 EEG and peripheral physiological signals. **Downsampling and Filtering**.
The physiological data in two datasets were downsampled to 128Hz. A bandpass frequency filter from 4.0-45.0Hz was applied for EEG. GSR was calculated and low-pass filtered with a 60Hz cut-off frequency in the AMIGOS dataset. In the DEAP dataset, no filter was processed for GSR.

**Region of Interest Finding**. In two datasets, the physiological signal's ROI corresponds to the ROI in the related face video, i.e., we take the physiological signal within 4s adjacent to the apex frame.

### 3.1.7 Imbalanced Data Balancing.

**AMIGOS Dataset**. We relabeled the valence and arousal scores greater than 0 as high, and scores less than 0 as low. The total number of low and high classes for all participants' trials for valence were 1784 and 510, and for arousal, 1980 and 314 in the short video experiment, and the total number of low and high classes for all participants' trials for valence were 3097 and 949, and for arousal, 3312 and 734 in the long video experiment. As can be seen, both datasets were not balanced among classes. To solve the long tail effect, we finally selected 1706 samples that could satisfy the total number of low and high classes for all participants' trials for valence and arousal were equal.

**DEAP Dataset**. We relabeled the valence and arousal scores greater than 5 as high, and scores less than or equal to 5 as low. The total number of low and high classes for all participants' trials for valence were 409 and 471, and for arousal, 367 and 513 in the experiment. As can be seen, both datasets were not balanced among classes. To solve the long tail effect, we finally selected 680 samples that could satisfy the total number of low and high classes for all participants' trials for valence and arousal were equal.

## 3.2 Feature Encoding Modules

### 3.2.1 Half Face Encoding Module.

As depicted in Figure 3, we used a 3D VGG-like backbone to extract spatial-temporal features for every frame from 100 input frames (size: $320 \times 320$) around each apex frame. The 3D VGG-like backbone consists of five layers. It takes the 100 frames from the ROI as input and outputs $F_{half-face} \in \mathbb{R}^{61 \times 128}$.

### 3.2.2 Physiological Signal-Aware Module.

To enhance emotion recognition operations for affective computing tasks applied to physiological signals, the multi-modal EmotionNet model has been introduced [26] (see Figure 3). This model is designed to capture both the heterogeneity and interactivity between EEG and GSR signals. It features a dual-stream transformer structure and an Interactivity-based Modal Fusion (IMF) module.

The **dual-stream transformer structure** comprises two convolutional blocks (EEG Filter and GSR Filter) followed by transformer encoders (TEs). These TEs employ a multi-head attention mechanism to capture dependencies between elements of the input feature maps. The mechanism is defined as:

$$\text{Multi-Head}(Q, K, V) = \text{Concat}(\text{head}_1, \ldots, \text{head}_h) \tag{5}$$

where $Q$, $K$, and $V$ are the query, key, and value matrices, and hh is the number of heads. Each head is computed as:

$$\text{head}_i = \text{Attention}(QW_i^Q, KW_i^K, VW_i^V) \tag{6}$$

where $W_i$ represent the weights matrices of $Q$, $K$, and $V$, respectively. The output is then concatenated and projected back to the original dimensionality.

The **Interactivity-based Modal Fusion Module** fuses features from both modalities based on interactivity scores. It is defined as:

$$\text{IMF}_{Output} = \text{IE}_1(X_1) \oplus \text{IE}_2(X_2) \tag{7}$$

where $X_1$ and $X_2$ are the input features from different modalities, and $\oplus$ denotes element-wise summation. The Interactivity Extractor (IE) generates interactivity scores from the concatenated features of different modalities. The interactivity extraction module achieves the following:

$$\text{IE}_{Output} = \text{ReLU}(\text{FC}_2(\text{ReLU}(\text{FC}_1(\text{IE}_{input})))) \odot \text{IE}_{input} \tag{8}$$

where $\text{FC}_1$ and $\text{FC}_2$ are fully connected layers, ReLU is the rectified linear unit activation function, and $\odot$ denotes the Hadamard product (element-wise multiplication) between the interactivity score and the input feature $IE_{input}$.

The result obtained after merging EEG and GSR is $F_{physio} \in \mathbb{R}^{61 \times 128}$.

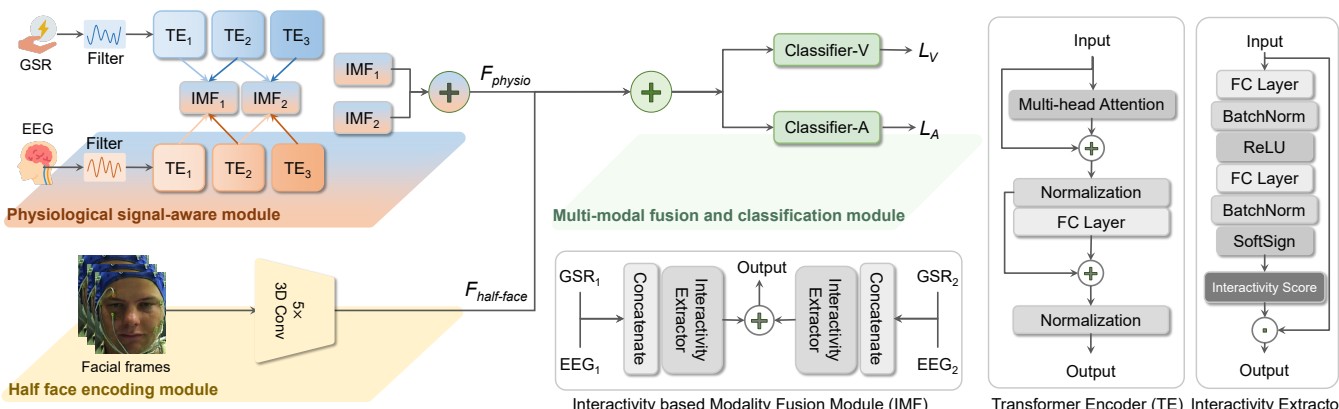

**Figure 3: The proposed method's architecture uses facial frames around the apex frame and time series data from two physiological signals (EEG and GSR) as input. These inputs are processed by a physiological signal-aware module and a half face-encoding module to extract comprehensive semantic features from different modalities. A multi-modal fusion and multi-task module is then employed to integrate these multi-branch features, ultimately predicting valence and arousal.**

### 3.3 Multi-Modal Fusion and Multi-task Module

We concatenate the half-face features $F_{half-face}$ and the physiological features $F_{physio}$. The concatenated output is then sent to two classification layers, which predict the emotional dimensions of valence and arousal (See Figure 3).

### 3.4 Loss Function

In multitask learning, we simultaneously predict the binary classification problem of valence and arousal. For each task, we use the cross-entropy loss function to measure the difference between the model's predictions and the true labels. Specifically, the loss functions $L_V$ and $L_A$ for valence and arousal can be defined as follows:

$$L_V = -\frac{1}{N} \sum_{i=1}^{N} y_{V_i} \log(p_{V_i}) + (1 - y_{V_i}) \log(1 - p_{V_i})$$

$$L_A = -\frac{1}{N} \sum_{i=1}^{N} y_{A_i} \log(p_{A_i}) + (1 - y_{A_i}) \log(1 - p_{A_i})$$

(9)

where $N$ is the number of samples, $y_{V_i}$ and $y_{A_i}$ are the true labels of the $i$-th sample for valence and arousal, respectively, and $p_{V_i}$ and $p_{A_i}$ are the predicted probabilities of the $i$-th sample being in the positive class for valence and arousal, respectively.

During training, the model tries to minimize the total loss $L = L_V + L_A$, which is the sum of the cross-entropy losses for each task. Note that this is a simple example, and in a real-world scenario, you might want to consider weighting the losses based on the importance of each task.

### 3.5 Test Set

To evaluate the effectiveness of our method, we conducted a real-world test set by inviting 10 participants complete experiments for the facial expression recognition task under partial occlusion from XR headsets. This section outlines the test set creation process.

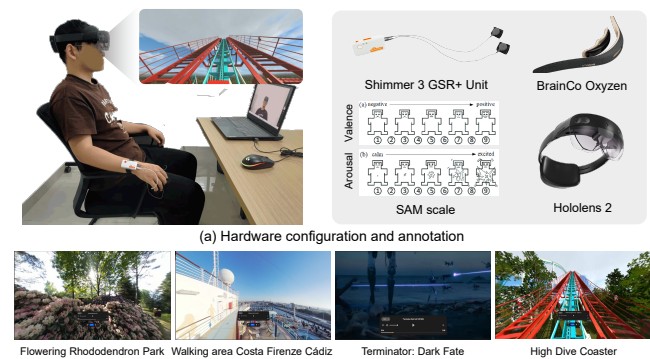

(a) Hardware configuration and annotation

Flowering Rhododendron Park   Walking area Costa Firenze Cádiz   Terminator: Dark Fate   High Dive Coaster

(b) 3D VR videos

**Figure 4: Generation procedure of real-world test set.**

The test set creation process, shown in Figure 4, includes six components divided into hardware and software categories, and two questions from the SAM questionnaire [2] on valence and arousal.

#### 3.5.1 Hardware.

The **BrainCo Oxyzen portable EEG device**[2] records EEG data at a sampling rate of 256Hz. It's designed based on the voltage difference between two forehead electrodes and weighs about 57g.

The **Shimmer 3 GSR+ Unit Sensing device**[3] aptures GSR signals at 128Hz and weighs about 30g.

The **Hololens 2 MR HMD**[4] displays the AR environment and allows interactions. It offers a resolution of 1440 × 936 per eye, a refresh rate of 60 FPS, and a FoV of 29°/43° (vertical/horizontal). It weighs about 566g and is compatible with the BrainCo Oxyzen EEG device, which we attach to the inside of the HoloLens lens.

---

[2]https://www.brainco-hz.com/
[3]https://www.shimmersensing.com/
[4]https://www.microsoft.com/en-us/hololens/buy

| Methods | Dataset | | Modality | | | Accuracy | | F1-score | | Inference Time (ms) | |
|---|---|---|---|---|---|---|---|---|---|---|---|
| | A | D | FF | HF | P | Valence | Arousal | Valence | Arousal | Valence | Arousal |
| Face Input Comparison: Apex and adjacent frames vs Apex frame only | | | | | | | | | | | |
| ResNet50 [15] (Single frame) | ✓ | ✗ | ✗ | ✓ | ✗ | 87.39 | 87.97 | 86.43 | 87.14 | 7.45 | 7.45 |
| Ours (Continuous frames) | ✓ | ✗ | ✗ | ✓ | ✓ | **92.37** | 92.08 | **91.92** | 91.74 | 11.73 | |
| ResNet50 [15] (Single frame) | ✗ | ✓ | ✗ | ✓ | ✗ | 53.67 | 60.29 | 46.15 | 60.29 | 7.37 | 7.37 |
| Ours (Continuous frames) | ✗ | ✓ | ✗ | ✓ | ✓ | 56.61 | **66.91** | 62.42 | **62.18** | 12.36 | |
| Face Input Comparison: Full face vs Half face & Physiological data | | | | | | | | | | | |
| Full face | ✓ | ✗ | ✓ | ✗ | ✗ | 92.08 | **92.96** | 91.64 | **92.81** | 2.18 | |
| Ours | ✓ | ✗ | ✗ | ✓ | ✓ | **92.37** | 92.08 | **91.92** | 91.74 | 11.73 | |
| Full Face | ✗ | ✓ | ✓ | ✗ | ✗ | **57.35** | 65.44 | **63.75** | 60.50 | 2.85 | |
| Ours | ✗ | ✓ | ✗ | ✓ | ✓ | 56.61 | **66.91** | 62.42 | **62.18** | 12.36 | |
| Ablation study | | | | | | | | | | | |
| FERPO$_{half-face-only}$ | ✓ | ✗ | ✗ | ✓ | ✗ | 87.68 | 90.61 | 87.71 | 90.06 | 2.46 | |
| FERPO$_{physio-only}$ | ✓ | ✗ | ✗ | ✗ | ✓ | 74.78 | 73.60 | 72.61 | 75.54 | 10.57 | |
| FERPO$_{full}$ (Ours) | ✓ | ✗ | ✗ | ✓ | ✓ | **92.37** | 92.08 | **91.92** | 91.74 | 11.73 | |
| FERPO$_{half-face-only}$ | ✗ | ✓ | ✗ | ✓ | ✗ | 52.20 | 61.76 | 51.12 | 59.37 | 2.54 | |
| FERPO$_{physio-only}$ | ✗ | ✓ | ✗ | ✗ | ✓ | 52.20 | 52.94 | 50.38 | 40.74 | 10.78 | |
| FERPO$_{full}$ (Ours) | ✗ | ✓ | ✗ | ✓ | ✓ | 56.61 | **66.91** | 62.42 | **62.18** | 12.36 | |
| Test set | | | | | | | | | | | |
| FERPO$_{full}$ (Ours) | ✗ | ✗ | ✗ | ✓ | ✓ | **61.33** | **54.67** | **65.06** | **66.00** | 11.70 | |

**Table 2: Comparasion of facial expression recognition performances on AMIGOS (A) and DEAP (D) dataset. Note: Full Face (FF), half Face (HF), physiological signal (P), and facial expression recognition performance under partial occlusion (FERPO).**

We use the **camera of Lenovo Legion R7000P** to capture the occluded face when wearing the HMD Hololens 2. The camera operates at a resolution of 1280 × 720 and a frame rate of 30 fps.

### 3.5.2 Software.

The **Python application for BrainCo Oxyzen** streams and collects EEG data. The complete code is available in the official open-source GitHub repositories of BrainCo [5].

The **Python application for Shimmer** streams and collects GSR data. The full code will be open-sourced.

### 3.5.3 3D Emotional Inducing.

Emotions and senses induced by 2D display may differ from those by the three-dimensional (3D) real world [17, 29, 46, 47]. Therefore, to verify the generalizability of our method, we use 3D video as mood induction procedures (MIPs).

### 3.5.4 Data collection.

We invited 10 participants (age: 22-29, 2 females) to watch four 3D VR videos from DEO VR [6] each lasting about 3 minutes. While watching the videos, we asked the participants to avoid large head movements. At the same time, we used BrainCo Oxyzen and Shimmer 3 GSR+ Unit to synchronously collect the participants' EEG and GSR signals. We eventually obtained about 1 hours of data.

### 3.5.5 Annotation.

We use two questions in the SAM questionnaire of valence and arousal, and ask participants to complete the questions to annotate their valence and arousal after watching the 3D videos. We finally obtained about 150 test set samples.

## 4 EXPERIMENT

### 4.1 Dataset Split

In our experimental evaluation, we conduct tests on the AMIGOS (macro-expression) [30] and DEAP (micro-expression) [21] datasets. Following the data preprocessing guidance [21, 30] and a similar protocol to [36], we split the dataset into training and validation sets in an 8:2 ratio Each sample includes consecutive frames around the apex frame, as well as EEG and GSR signals.

---

[5]https://github.com/BrainCoTech/oxyzen-example

[6]https://deovr.com/

## 4.2 Baseline

We carefully establish the baselines by comparing our novel method with state-of-the-art methods on the facial expression recognition task under partial occlusion from XR headsets. Experiments are performed with the Pytorch framework on an NVIDIA 4090 GPU.

## 4.3 Metric

Following the standard evaluation metric for facial expression recognition tasks, we employ three commonly used metrics, namely accuracy, F1-score, and Prediction time to measure the performance of our method for both valence prediction and arousal prediction. Additionally, we report the "time cost of prediction in the validation set" as the model may be applied to real-time emotion recognition of HMD in XR applications.

## 4.4 Experiment on the Validation Set

### 4.4.1 Apex and adjacent frames vs Apex frame only.

Our method's accuracy, F1-score surpass the method [15] of facial expression under partial occlusion, which uses a single frame as input, by 5.69%, and 6.35% on AMIGOS dataset and 5.47%, and 10.98% on DEAP dataset, respectively, (see Table 2). Moreover, the inference time is suitable for real-time tasks. This is attributed to our method's superior handling of occluded face information and the incorporation of physiological signals. The continuous signal frames capture more temporal information, mitigating the impact of single frame noise or detail loss. Our approach not only enhances facial expression recognition performance under partial occlusion (FERPO) for XR HMDs but also reduces prediction time, making it ideal for real-time XR applications. Furthermore, our multi-task approach offers benefits over single-task methods, including fewer parameters, faster convergence, and improved performance. Additionally, Micro-expression datasets underperformed in tests compared to macro-expressions, likely due to their subtler facial changes. Thus, for HMD users, merging physiological signals with facial data enhances emotion recognition when micro-expressions occur.

### 4.4.2 Full face vs Half face and physiological data.

Using half-face images and physiological data for simultaneous valence and arousal prediction yields higher accuracy, improved F1-score, and faster results than full-face images. This implies that the fusion of half-face and physiological signals achieves comparable emotion prediction accuracy to full-face images.

## 4.5 Ablation Study

In this subsection, we aim to conduct a detailed analysis of the contributions of visual and physiological modalities in our proposed approach. Table 2 shows that $FERPO_{physio-only}$ performs significantly worse than both $FERPO_{half-face-only}$ and $FERPO_{full}$ on the AMIGOS and DEAP datasets. This is because $FERPO_{physio-only}$ only uses physiological data and ignores facial information that provides details about the subject's emotional expressions. As a result, when physiological signals alone may not accurately determine the emotional state, it leads to inaccurate valence and arousal outcomes. This highlights the importance of facial information in improving the accuracy of emotional state prediction. Furthermore, $FERPO_{half-face-only}$ also performs significantly worse than

$FERPO_{full}$. This is because $FERPO_{half-face-only}$ only uses half-face details and ignores the spatial relationship of the complete facial key points. Consequently, this greatly increases the difficulty of predicting positive emotions, negative emotions, and arousal levels, leading to more imprecise valence and arousal outcomes.

This underlines the importance of supplementing additional physiological modalities for resolving the incomplete information problem caused by partial occlusion. The inclusion of physiological information greatly enhances the model's ability to accurately identify complex emotional states.

## 4.6 Experiment on the Real-world Test Set

The test set experiment demonstrates the effectiveness of our method in handling complex real-world scenarios where participants use actual HMDs, and has a extremely low inference time. This implies that our method may be able to fulfill the task of real-time emotion recognition under the use of HMDs, such as in VR Therapy [6, 18, 31, 41], VR Relaxation [14], AR education [12, 23].

## 5 LIMITATIONS AND FUTURE WORK

The datasets we used, AMIGOS and DEAP, use non-immersive two-dimensional (2D) videos or images as mood induction procedures (MIPs), such as images, audios, and videos. However, the emotion and sense induced by 2D display are different from that by the three-dimensional (3D) real world; for example, the 2D display lacks the sense of presence and depth information [47]. Moreover, considering the known differences in EEG dynamics between 2D and 3D presentations [17], there may be a gap between the experimental studies based on 2D display and real-world applications [17]. Therefore, we plan to introduce 3D extended reality (XR) MIPs, which enable researchers to simulate real-world conditions in a controlled laboratory environment, and are gaining popularity in studying emotion.

Additionally, the facial images collected after the subject wears an HMD may be affected by interference factors such as lighting, head shaking or deflection. The facial detection algorithm currently used is sensitive to disturbances such as blurred images and weak lighting. In the future, we will explore more robust and stable facial detection algorithms.

## 6 CONCLUSION

We've introduced a novel multi-task method for emotion recognition in XR, using physiological signals and occluded facial expressions. Our approach integrates EEG and GSR data with half-face expressions via a feature-level fusion, enabling simultaneous prediction of valence and arousal. Our approach notably enhances emotion recognition accuracy under partial occlusion. This improvement is evident in AMIGOS and DEAP datasets, as well as in real-world situations, and it's validated for use in emotion recognition tasks in HMD-based XR. Future enhancements will focus on robustness against disturbances like lighting and head movements, and the application of 3D XR mood induction procedures for a more immersive experience. Our work significantly advances emotion recognition in XR, setting the stage for more responsive virtual environments.

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
