# OpenReview forum: "Emotion Recognition in HMDs: A Multi-task Approach Using Physiological Signals and Occluded Faces"
_acmmm.org/ACMMM/2024/Conference — MM2024 Poster_

### Official Review · Reviewer_74MT · 2024-05-03

**Rating:** 5
**Confidence:** 3

**Summary:**

In this study, the authors address the challenge of facial expression occlusion by HMDs in XR environments by integrating physiological signals (EEG and GSR) with partially visible facial expressions for improved emotion recognition. Using a multi-task approach featuring feature-level fusion, the proposed model enhances emotion detection under occlusion, demonstrating improvements in accuracy for half-face emotion recognition.

**Strengths:**

The field of study is in the scope of the ACMMM and is highly crucial in XR, and the manuscript is well-written and easy to follow. The proposed approach is novel enough for the publication.

**Limitations:**

There are some minor changes which are necessary to strengthen the claims and findings:


Have the authors explored the use of HMDs equipped with eye and gaze-tracking capabilities? It would be beneficial to discuss this. Additionally, the role of eye contact and the significance of eye features in FER should also be addressed.

1:
“We tested our multi-modal, multi-task method on DEAP, AMIGOS datasets, and real-world situations:” needs citations here.

The term “multi-task method” needs explanation in the Introduction, before summarizing the contributions.

2.1:
“The discrete model includes six universally recognized emotions”: eight classes of emotions also should be mentioned including neutral and contempt.

Please use abbreviations instead of the full form after the first appearance of each term throughout the text, like electroencephalography (EEG) in 2.3 or XR (Mixed Reality) in 2.4.

2.3:
CNN: mention the full form.

Table 1: needs more explanation and comparison in the text ( in 2.4).

3:
The transition to the METHODOLOGY section is quite abrupt; please provide an introductory paragraph to smooth into this section.

Figure 2: The caption needs more explanation, not informative.

3.1:
The title may better be updated, Data “Processing” Module, do you mean preprocessing?

3.1.2:
“Following previous research”: not clear how you defined the yellow boxes.

Figure 2(c) and (d) should be discussed in the text.

3.2.2:
“where 𝑄, 𝐾, and 𝑉 are the query, key, and value matrices, and hh”: correct the typo.

Why R61×128?


Figure 4: caption needs more info. Also, discuss it accordingly in the text.


Table 2: discuss clearly why the accuracy and F1 score in the D dataset are much lower compared to the A dataset in 4.4.1 ("Additionally, Micro-expression datasets underperformed in tests compared to macro-expressions, likely due to their subtler facial changes").

3.5.3: “MIPs” need more explanation.

3.5.4: include more details of participants' demography.

4.2: baselines and comparison models should be specifically defined here.

**Suitability:**

2

---

### Official Review · Reviewer_LrrN · 2024-05-24

**Rating:** 3
**Confidence:** 4

**Summary:**

The authors have implemented a recognition method that adopts a fusion approach of occluded facial expressions and physiological signals, achieving recognition performance surpassing that of  full-face images. This study proposes a new method to recognize emotion in XR that addresses the limitations of face occlusion. The results of this study show that for the half-face emotion recognition task, the introduction of physiological signals can significantly improve the performance of the model.

**Strengths:**

The paper is novel in the field of emotion recognition in XR by realizing feature fusion of visual images and physiological signals. Adequate experiments were also conducted to verify the effectiveness and advancedness of the method. Furthermore, the paper constructs data from real scenarios and tests the effectiveness and performance of the proposed method on real-world situations, and the results perform well.

**Limitations:**

1. The novelty of this work is not clear, and the authors must clarify their innovations. I suggest the authors rewrite the introduction.
2. On line 153 of the introduction section, if the recognition method combining occluded images and physiological signals is not proposed by the authors for the first time, it should be explicitly stated in the introduction.
3. Many details of the method are not provided, such as network details, training details, etc.
4. The experimental section lacks comparison with peer works, especially with benchmark works.
5. Additional ablation experiments are needed to separately demonstrate the effectiveness of the multi-task method compared to the single-task method.

**Suitability:**

3

---

### Official Review · Reviewer_iVbG · 2024-05-31

**Rating:** 3
**Confidence:** 3

**Summary:**

The paper titled "Emotion Recognition in HMDs: A Multi-task Approach Using Physiological Signals and Occluded Faces" addresses the challenge of recognizing emotions in Extended Reality (XR) environments, where Head-Mounted Displays (HMDs) partially occlude facial expressions, thereby hindering accurate Facial Expression Recognition (FER). The authors propose a multi-task model that integrates physiological signals (Electroencephalography (EEG) and Galvanic Skin Response (GSR)) with partially visible facial expressions to predict emotional states in terms of valence and arousal. By employing feature-level fusion and temporal analysis of physiological signals, the study demonstrates improved accuracy in emotion recognition under partial occlusion conditions. The model is validated on the DEAP and AMIGOS datasets, showing significant performance enhancements.

**Strengths:**

1、The study introduces a novel multi-task model that effectively combines physiological signals with occluded facial expressions, addressing a critical challenge in XR environments.
2、The model is rigorously tested on established datasets (DEAP and AMIGOS) and real-world scenarios, demonstrating its robustness and applicability.
3、The integration of EEG and GSR signals with facial expressions significantly enhances the accuracy of emotion recognition, especially under conditions of partial facial occlusion.
4、The inclusion of temporal physiological signals in the model provides a deeper understanding of underlying emotions and improves recognition performance.
5、The research has practical implications for real-time applications in social XR, virtual therapy, and other immersive environments where understanding user emotions is crucial.

**Limitations:**

1、The model's performance is highly dependent on the quality and variety of the datasets used for training. Real-world applicability might vary with different user demographics and contexts not covered by the DEAP and AMIGOS datasets.
2、The multi-task model and the integration of multiple signals increase the computational complexity, which may pose challenges for real-time processing and scalability in practical applications.
3、Physiological signals like EEG and GSR can be noisy and subject to individual differences, which might affect the model's generalizability across different users.
4、While the study focuses on partially occluded faces, it primarily considers the lower face. The model might not perform as well if other regions are occluded or if the occlusion pattern changes dynamically.
5、The feature-level fusion method, although effective, might not fully capture the complex interactions between different modalities.
6、In the experimental section, the performance comparison only involves one method which is from 2020. It is suggested to incorporate comparisons with multiple state-of-the-art methods to establish the advancement of the proposed approach.
7、The tense of the article is confusing.
8、There are some punctuation errors in the text. For example, some double quotation marks in the text are Chinese symbols.
9、The title of Table 2 should be positioned above the table.
10、Some parts of the article are incorrectly expressed. For example, the function f below Equation 2 should be “Function f(.)” rather than “Function (f )”.
11、There are some grammatical errors in the text. For example, below equation 6 in Section 3.2.2 “where  represent the weights matrices of Q, K, and V, respectively.”, “represent” should be “represents”.

**Suitability:**

2

---

### Official Review · Reviewer_d8Ec · 2024-05-31

**Rating:** 5
**Confidence:** 4

**Summary:**

It is about the emotion recognition in XR. Since the HMD occlude people's faces, the tradition facial expression recognition methods are not applicable in this context. Thus, it proposes a multi-modal emotion recognition method which incorporates EEG and GSR.

**Strengths:**

It uses multi-modal and multi-task emotion recognition method to solve the occlusion problem in XR based facial expression recognition. The research view is novel.

**Limitations:**

It is better to perform more comprehensive benchmark test.

**Suitability:**

3

---

### Meta-Review · Area_Chair_eH6n · 2024-07-01

**Recommendation:** Accept (Poster)
**Confidence:** 4

**Metareview:**

This paper proposes a novel method for emotion recognition in XR environments that addresses the challenge of facial occlusion caused by Head-Mounted Displays (HMDs). The method integrates physiological signals (EEG and GSR) with partially visible facial expressions using a multi-task approach with feature-level fusion. While all reviewers acknowledged the novelty of the approach and its potential significance for XR, they also raised several limitations that need to be addressed.

**Reviewer Consensus:**

* Strengths:
    * **Novelty:** All reviewers agreed that the combination of physiological signals and facial expressions for emotion recognition in XR is a novel approach.
    * **Improved Accuracy:** The proposed method achieves better accuracy compared to using only facial expressions, especially under occlusion conditions.
    * **Rigorous Testing:** The model is validated on established datasets (DEAP and AMIGOS) and demonstrates its effectiveness.
* Weaknesses:
    * **Limited Comparisons:** Several reviewers pointed out the lack of comparisons with state-of-the-art methods and the need for ablation studies to isolate the effectiveness of the multi-task approach.
    * **Missing Details:** Reviewers requested more details about the network architecture, training procedures, and data processing steps.
    * **Unclear Writing:** Minor grammatical errors and inconsistencies were identified.

Considering the overall reviewer sentiment and the paper's potential contribution to the field, I recommend accepting this submission.